# Minimum Variance Unbiased N:M Sparsity for the Neural Gradients

**Brian Chmiel** [†○*] **Itay Hubara** [†○*] **Ron Banner** [†] **Daniel Soudry** [○]

[†]Habana Labs – An Intel company, Caesarea, Israel,
[○]Department of Electrical Engineering - Technion, Haifa, Israel

{bchmiel, ihubara, rbanner}@habana.ai
daniel.soudry@gmail.com

## Abstract

In deep learning, fine-grained N:M sparsity reduces the data footprint and bandwidth of a General Matrix multiply (GEMM) up to x2, and doubles throughput by skipping computation of zero values. So far, it was mainly only used to prune weights to accelerate the forward and backward phases. We examine how this method can be used also for the neural gradients (i.e., loss gradients with respect to the intermediate neural layer outputs). To this end, we first establish a tensor-level optimality criteria. Previous works aimed to minimize the mean-square-error (MSE) of each pruned block. We show that while minimization of the MSE works fine for pruning the weights and activations, it catastrophically fails for the neural gradients. Instead, we show that accurate pruning of the neural gradients requires an unbiased minimum-variance pruning mask. We design such specialized masks, and find that in most cases, 1:2 sparsity is sufficient for training, and 2:4 sparsity is usually enough when this is not the case. Further, we suggest combining several such methods together in order to potentially speed up training even more. A reference implementation is supplied in the supplementary material.

## 1 Introduction

Pruning Deep Neural Networks (DNNs) is one of the most effective and widely studied methods to improve DNN resource efficiency. Since DNNs are over-parametrized, most researchers focused on weights pruning. Yet, recently researchers suggested that sparsity of activations (Jaszczur et al., 2021; Kurtz et al., 2020) and gradients (Chmiel et al., 2021b) could be exploited as well. However, all these types of *unstructured* pruning only reduce the memory footprint (Frankle & Carbin, 2018; Evci et al., 2020). It is possible to also reduce the compute footprint by enforcing some structure on the pruning mask, such as block sparsity (Wen et al., 2016), filter sparsity (Li et al., 2017), or N:M fine-grained sparsity (Nvidia, 2020; Hubara et al., 2021; Mishra et al., 2021).

We focus on N:M fine-grained sparsity, in which, N out of every M contiguous elements would be pruned, for at least one of the two matrices involved in the matrix multiplication. Nvidia's sparse tensor cores (Nvidia, 2020; Mishra et al., 2021) can use N:M fine-grained sparsity to accelerate matrix multiplication. Specifically, Nvidia (2020) used a 2:4 format to accelerate inference up to x2. They suggested using a three-step scheme: (a) train a dense model, (b) prune weights to obtain a 2:4 fixed mask, and (c) use the original training regime to retrain with the masked weights.

Following works suggested methods to accelerate different parts of this scheme. First, Zhou et al. (2021) was able to omit steps (a) and (b) by training with an N:M mask from scratch using a straight-through estimator (STE) and additional regularization. Specifically, they keep a dense copy of the weights and set different weight decays rates to the masked and unmasked weights. Next, Hubara et al. (2021) focused on accelerating the remaining step (c), i.e., sparse training, as we do here.

---

[*]Equal contribution.

Table 1: Exploring fine-grained sparsity on different training phases with different sampling methods (MVUE, MSE). While previous methods aim to accelerate the forward and backward phases, we focus on accelerating the update phase. The combination of all methods allows us to accelerate all training phases.

| Phase | Tensor | | | |
|---|---|---|---|---|
| | Weights (Nvidia, 2020) | T-Weights (Hubara et al., 2021) | Gradients (**ours**) | Training T-Weights + Gradients |
| Forward | ✓(MSE) | ✓(MSE) | ✗ | ✓(MSE) |
| Backward | ✗ | ✓(MSE) | ✗ | ✓(MSE) |
| Update | ✗ | ✗ | ✓(MVUE) | ✓(MVUE) |

Recall that in each training step we use Backpropagation, which has three phases. Generally, each phase requires a General Matrix Multiplication (GEMM) for each DNN layer $l$:

$$[\textbf{Forward}] \quad z_l = W_l h_{l-1}; \qquad h_l = f_l(z_l) \tag{1}$$

$$[\textbf{Backward}] \quad g_l = \text{Diag}(f'_l(z_l))W^T_{l+1}g_{l+1} \tag{2}$$

$$[\textbf{Update}] \quad \frac{\partial C}{\partial W_l} = g_l h^T_{l-1}, \tag{3}$$

where $C$ is the loss function, and in each layer $l$, $f_l$ is a non-linear activation function, $W_l$ represents the weights, $z_l$ the pre-activations, $h_l$ the post-activations and $g_l = \frac{\partial C}{\partial z_l}$ is the neural gradient.

Nvidia suggested accelerating only the inference phase (i.e., the forward pass in eq. Equation (1)), while the backward and update passes were kept dense. Noting that the backward phase uses the transposed (sparse) weight matrix, Hubara et al. (2021) used a transposable mask, i.e., a mask that can be transposed and still match the N:M fine-grained structure. This enabled the acceleration of the backward phase. Although Hubara et al. (2021) suggested different methods to find the optimal transposable mask efficiently, they did not suggest how to accelerate the update phase.

In this work we explore different methods to accelerate the update phase as well using N:M sparsity. We need to decide in Equation (3) if we want to prune the activations ($h_l$) or the neural gradients ($g_l$). In order to avoid a mismatch with the forward phase in Equation (1), where the activations are not pruned, we decided in this work to focus on the neural gradient for the update phase. To that end, we examine gradients with fine-grained pruning and establish a tensor-level optimality criteria. So far, N:M sparsity in the weights was obtained by minimizing the Mean Square Error (MSE). We explain (Section 3) that, while this MSE criterion can also be used for the N:M sparsity in activations (which can be useful for inference, as we discuss in Section 6), for N:M sparsity in the neural gradients it is better to use a Minimum Variance Unbiased Estimate (MVUE).

We develop (in Section 4) such MVUE pruning methods for 1:2 and 2:4 sparsity in the neural gradients. Our experiments (in Section 5) suggest that while the traditional minimum MSE method crashed, our MVUE method with 1:2 sparsity is usually sufficient for training, and 2:4 sparsity is enough when this is not the case. Moreover, we suggest to combine several such methods together (fine-grained sparse neural gradients and sparse transposable fine-grained weights) in order to potentially speed up training even more and be able to accelerate all training phases with N:M fine-grained sparsity. In Table 1 we present all the N:M fine-grained structured sparsity methods, which part of the network they accelerate, the relevant optimality criteria we use, and the configurations we use to fully accelerate training.

In summary, this paper makes the following contributions:

• We developed an unbiased minimum variance optimality criteria for pruning neural gradients with N:M structured sparsity.
• We propose 1:2 and 2:4 unbiased minimum variance methods to prune the neural gradients and demonstrate that they achieve small or no degradation, where previous methods failed.
• We combine these methods with previous methods for N:M structured sparsity in the weights, and observe small or no degradation. Thus, the GEMMs in all training phases can potentially be accelerated by x2.

## 2    RELATED WORKS

Pruning has been extensively investigated in the last few years. Most of the pruning methods focus on pruning the weights (Evci et al., 2020; Frankle & Carbin, 2018; Janowsky, 1989; Liu et al., 2018). Unstructured pruning methods achieved impressive sparsity ratio with minimal or no accuracy degradation, e.g. Renda et al. (2020) achieved over 80% sparsity in ResNet50 over the ImageNet dataset without sacrificing accuracy. Despite this impressive achievement, the ability of unstructured pruning methods to actually reduce computational resources of modern hardware is limited (Nvidia, 2020; Mishra et al., 2021).

Structured pruning methods vary between coarse-grained and fine-grained methods. Coarse-grained methods such as filter-wise or layer-wise pruning (Li et al., 2017; Luo et al., 2017; Wen et al., 2016) are naturally supported by hardware and software but these methods were only able to maintain the test accuracy for sparsity ratio significantly lower than 50%. Recently, Nvidia introduced the Ampere GPU architecture (Nvidia, 2020; Mishra et al., 2021) hardware with software support for N:M fine-grained structured sparsity. Specifically, they showed that 2:4 fine-grained structured sparsity, where two of every four contiguous elements are zero, achieves up to x2 improvement in the GEMM operation. They suggested a three-step scheme to accelerate inference. Later, Zhou et al. (2021) accelerated their method by avoiding the first two steps. Next, Hubara et al. (2021) accelerated the remaining training step by suggesting transposable mask, which accelerates both the forward and backward phases ($\frac{2}{3}$ of the training). Stosic & Stosic (2021) further demonstrated the transposable mask can accelerate training with minimal accuracy degradation on 20 different models for various tasks and datasets. Pool & Yu (2021) suggested permuting the weight matrices to improve accuracy of sparse models for inference. Sun et al. (2021) suggested a mixed layer-wise N:M sparsity schemes to improve the uniform sparsity scheme with similar complexity constraints. Holmes et al. (2021) suggests a new learning framework to improve the performance of N:M sparse NLP models on downstream tasks.

Beyond pruning the weights, recent work also focuses on unstructured sparsity of the activations or neural gradients. Kurtz et al. (2020) suggested a parametrized activations function called Forced-Activation-Threshold Rectified Linear Unit (FATReLU) which increases the naturally sparse of ReLU with any accuracy loss. Jaszczur et al. (2021) studied the sparsification of the activations in Transfomer-based models. "MeProp" (Sun et al., 2017) prunes the K smallest absolute-valued entries of the neural gradients on the fly, using the top-k algorithm. Aamir Raihan & Aamodt (2020) used top-k pruning on the copies of weights and activations used in the backpropagation. Ye et al. (2019), suggested "stochastic pruning", reaching higher sparsity levels on the neural gradient. Chmiel et al. (2021b) improved their results with a lognormal distribution approximation for the neural gradient achieving more than 80% sparsity on the neural gradients without accuracy degradation.

In parallel to our work, two additional works suggested to use N:M structured sparsity to be able to accelerate training: In the first, McDanel et al. (2022) suggested a method to use N:M structured data pruning for the neural gradients to accelerate the backward phase, which was also accelerated in Hubara et al. (2021). In Appendix B.3 we show the degradation of applying McDanel et al. (2022) method also in the update phase. In the second, Weixiang et al. (2022) suggested to use the spatial similarity in vision models to fix the loss of information after applying the N:M mask. Their mask is applied on the weights and activations, while keeping the neural gradients in full precison. However, this spatial similarity can not be exploited in other domains such as natural language processing. As far as we know, no previous work suggested using N:M fine-grained sparsity to accelerate the update phase, by pruning the neural gradients.

## 3    WHICH OPTIMALITY CRITERIA TO USE?

When pruning weights during training, we require a local (tensor level) criterion to select which weights to prune. A popular criterion is minimizing the Mean Square Error (MSE). For a deterministic vector $\mathbf{a}$ and a random pruning operator $\theta$ we can write the MSE of pruning as

$$\text{MSE}[\theta(\mathbf{a})] = E||\theta(\mathbf{a}) - \mathbf{a}||^2 = E||\theta(\mathbf{a}) - E\theta(\mathbf{a})||^2 + ||E[\theta(\mathbf{a})] - \mathbf{a}||^2 \triangleq \text{Var}[\theta(\mathbf{a})] + \text{Bias}^2[\theta(\mathbf{a})],$$

where $E$ denotes an expectation over the randomness of $\theta(\mathbf{a})$.

Recently, Chmiel et al. (2021a) investigated which optimality criteria to use, but in the context of quantization (i.e., there $\theta(\mathbf{a})$ was a quantizer). They found that, when quantizing the weights or activations, we should indeed minimize the MSE of the quantization error. In contrast, for the neural gradients, they found that it is critical to use unbiased quantization (i.e., $\text{Bias}[\theta(\mathbf{a})] = 0$) such as stochastic rounding. Specifically, Chmiel et al. (2021a) showed that unbiasedness in the neural gradient leads to an unbiased estimator of the weight mini-batch gradient, which enables proper convergence of SGD, according to standard SGD analysis (e.g. Bottou et al. (2018)). Therefore, we suggest to apply N:M fine-grained sparsity on the neural gradients using the same optimality criteria and focus on finding an unbiased estimator. From all the possible unbiased estimators, we will prefer the one that reduce the MSE. Since we focus on an unbiased estimator (i.e., $\text{Bias}[\theta(\mathbf{a})] = 0$), all that remains is to minimize the variance $\text{Var}[\theta(\mathbf{a})]$. Therefore, we conclude that the Minimum Variance Unbiased Estimator (MVUE) is optimal for the neural gradients.

## 4 MINIMUM VARIANCE UNBIASED ESTIMATOR FOR N:M SPARSITY

In this section, we propose two unbiased estimators with minimum variance: one for the 1:2 case and then another to the 2:4 case. Given a block of entries (i.e. a vector) $a$, each estimator produces another block $\theta(\mathbf{a})$ with the relevant $N : M$ sparsity pattern.

### 4.1 MINIMUM VARIANCE UNBIASED ESTIMATOR FOR 1:2 SPARSITY

For a block $\mathbf{a} \triangleq [a_1, a_2]$, one entry needs to be pruned, so any 1:2 method has the following form

$$\theta\left(\mathbf{a}\right) = \begin{cases} [v_1, 0] & \text{, w.p. } p \\ [0, v_2] & \text{, w.p. } 1 - p \end{cases}. \tag{4}$$

We wish to design an unbiased estimator for this pruning method, so

$$E[\theta\left(\mathbf{a}\right)] = [a_1, a_2]. \tag{5}$$

To find an unbiased estimator which minimizes the total block variance, using equations 4 and 5 we calculate the total variance of a block, as the sum of its element variances:

$$\text{Var}_B\left[\theta(\mathbf{a})\right] \triangleq \sum_i \text{Var}\left[\theta_i(\mathbf{a})\right] = \sum_i \left( E\left[\theta_i^2(\mathbf{a})\right] - E\left[\theta_i(\mathbf{a})\right]^2 \right) = v_1^2 p - a_1^2 + v_2^2(1-p) - a_2^2. \tag{6}$$

Using equations 4, 5, and 6 together we obtain the following expression for the total variance in the block, as a function of $v_1$ alone (for more information, see Appendix A.1):

$$\text{Var}_B[\theta(\mathbf{a})] = v_1 \cdot a_1 - a_1^2 + \frac{a_2^2 \cdot v_1}{v_1 - a_1} - a_2^2. \tag{7}$$

Since we wish to minimize this quantity, we differentiate equation 7 with respect to $v_1$, and equate to zero to obtain following unbiased estimator, which has the lowest variance of all unbiased estimators (full details in Appendix A.1):

$$\theta\left(\mathbf{a}\right) = \begin{cases} [\text{sign}(a_1) \cdot (|a_1| + |a_2|), 0] & \text{, w.p. } \frac{|a_1|}{|a_1| + |a_2|} \\ [0, \text{sign}(a_2) \cdot (|a_1| + |a_2|)] & \text{, w.p. } \frac{|a_2|}{|a_1| + |a_2|} \end{cases}. \tag{8}$$

Let us calculate the mean MSE of this unbiased method. By substituting into Equation 7 the optimal solution for $v_1$, the optimal estimator in Equation 8 has a variance of $2a_1 a_2$. Therefore, since the method is unbiased we obtain $\text{MSE} = \text{Bias}^2 + \text{Var} = 0 + 2a_1 a_2 = 2a_1 a_2$. In Figure 1 we present an illustration of the proposed MVUE 1:2. Table 2 compares different methods for 1:2 structured pruning of the neural gradients, on ResNet18 Cifar10 dataset. Notice, the proposed MVUE method has the best accuracy, although it does not minimize the MSE, as done by the 'greedy' method.

### 4.2 OPTIMALITY CRITERIA FOR 2:4

We now extend the results from the previous section to 2:4 pruning. With a block $a \triangleq [a_1, a_2, a_3, a_4]$, we construct an unbiased 2:4 block pruning method $\theta(\mathbf{a})$ with minimum variance. First, we note the method must satisfy the following condition

$$\theta_i(\mathbf{a}) = \frac{a_i}{p_i} \quad \text{with probability } p_i \tag{9}$$

Table 2: 1:2 sparsity on the neural gradients of ResNet18 cifar10 dataset. 'Greedy' is the traditional minimum MSE method of choosing the smallest element for each block. 'Biased' refers to the case $[v_1, v_2] = [a_1, a_2]$ in Equation (4) for $p = |a_1|/|a_1| + |a_2|$. 'Uniform' refers to uniform sample, i.e. $p = 0.5$, $[v_1, v_2] = [a_1, a_2]$. 'Unbiased' refers to unbiased uniform sampling, i.e $p = 0.5$, $[v_1, v_2] = [2a_1, 2a_2]$. 'MVUE' refers to the minimimum variance unbiased estimator in Equation (8).

| Method | Baseline | Greedy | Biased | Uniform | Unbiased | MVUE (Ours) |
|---|---|---|---|---|---|---|
| Accuracy (%) | 90.02 | 85.5 | 71.8 | 85.8 | 87.2 | **89.8** |

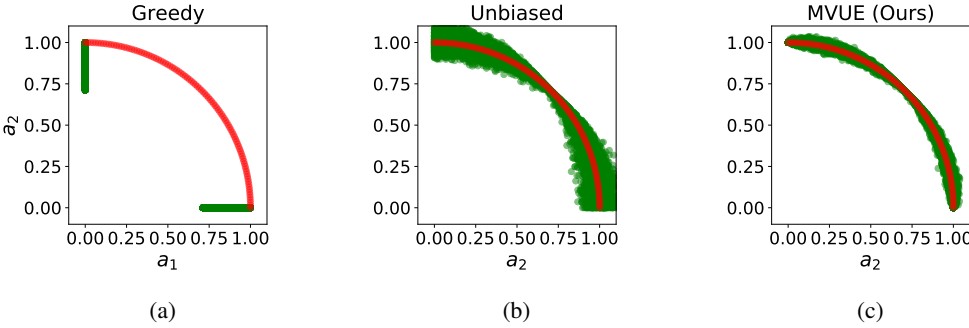

(a)  (b)  (c)

Figure 1: Fine-grained 1:2 Sparsity for blocks located on the first quarter of the unit circle. The blocks $[a_1, a_2]$ (represented by red dots) are sampled 100 times each, and then averaged (green dots) using one of three methods: (a) *greedy* is the traditional method that generates the block $[0, a_2]$ if $a_1 \leq a_2$, or $[a_1, 0]$ otherwise. In this method, all 100 samples are the same for each block, resulting in a biased average. (b) unbiased - each block $[a_1, a_2]$ is equally likely to be pruned to $[2a_1, 0]$ or $[0, 2a_2]$. Although the average of the 100 samples is unbiased, it does not have minimum variance. (c) Our unbiased method with minimum variance (Equation 8), has a smaller spread here than in (b).

since then, and only then, we get an unbiased estimate:

$$E[\theta_i(\mathbf{a})] = \frac{a_i}{p_i} \cdot p_i + 0 \cdot (1 - p_i) = a_i, \quad , \forall i \in \{1, 2, 3, 4\}. \tag{10}$$

In this case, the variance of each element in the pruned block is:

$$\text{Var}\left[\theta_i(\mathbf{a})\right] = E\left[\theta_i^2(\mathbf{a})\right] - E\left[\theta_i(\mathbf{a})\right]^2 = \left(\frac{a_i}{p_i}\right)^2 \cdot p_i + 0^2 \cdot (1 - p_i) - a^2 = \frac{a_i^2}{p_i} - a_i^2. \tag{11}$$

Then, the total variance in the pruned block is

$$\text{Var}_B\left[\theta(\mathbf{a})\right] \triangleq \sum_i \text{Var}\left[\theta_i(\mathbf{a})\right] = \sum_i \left(\frac{a_i^2}{p_i} - a_i^2\right). \tag{12}$$

We wish to minimize this quantity under the following equality and inequality constraints

$$\sum_i p_i - 2 = 0 \quad ; \quad p_i - 1 \leq 0 \ , \forall i \in \{1, 2, 3, 4\}. \tag{13}$$

Therefore, to find $p_i$, we need to apply the KKT conditions on the following Lagrangian:

$$L = \sum_j \left(\frac{a_j^2}{p_j} - a_j^2\right) + \sum_j \lambda_j (p_j - 1) + \mu \sum_j (p_j - 2). \tag{14}$$

Differentiating the Lagrangian with respect to $p_i$, we obtain

$$\frac{\partial L}{\partial p_i} = -\frac{a_i^2}{p_i^2} + \lambda_i + \mu = 0 \ , \forall i \in \{1, 2, 3, 4\}, \tag{15}$$

where, for each $i$, the constant $\lambda_i$ could be zero or positive. Using Equation 15 for the case $\lambda_i = 0$ we get that $p_i = a_i/\sqrt{\mu}$. This, coupled with the normalization constraint $(\sum_i p_i = 2)$ implies that

$$p_i = \frac{2a_i}{\sum_j a_j} \quad , \forall i \in \{1, 2, 3, 4\}. \tag{16}$$

Turning to the case where $\lambda_i > 0$ for some specific $i$, we have $p_i = 1$ because of the complementary slackness condition in KKT. The normalization constraint $(\sum_j p_j = 2)$ therefore guarantees that $\sum_{k \neq i} p_k = 1$. This implies all other $p_k$ (for $k \neq i$) are in the range $[0, 1]$, so the constraint $p_k \leq 1$ is slack, and therefore $\lambda_k = 0$ for every $k \neq i$. Therefore, from equation 15 we have that

$$\frac{\partial L}{\partial p_k} = -\frac{a_k^2}{p_k^2} + \mu = 0 \Rightarrow p_k = \frac{a_k}{\sqrt{\mu}} \quad , \forall k \neq i \tag{17}$$

Since $\sum_{k \neq i} p_k = 1$ we conclude that the optimality criterion is

$$\exists i : p_i = 1 \quad \text{and} \quad p_k = \frac{a_k}{\sum_{k \neq i} a_i} \quad , \forall k \neq i \tag{18}$$

Thus, a 2:4 fine-grained pruning method can be optimal only if it always satisfies either Equation 18 or 16. We provide such a method in Appendix A.2. This method allows us to sample pairs of elements for a 2:4 policy that always satisfies one of the criteria stated in Equations 18 and 16.

### 4.3  A COMPARISON OF THE OPTIMAL 1:2 AND OPTIMAL 2:4 METHODS

Given a block $a = [a_1, a_2, a_3, a_4]$, we can either apply optimal 2:4 method directly on that block $\theta_{2:4}(\mathbf{a})$ or we can break it into two sub-blocks $[a_1, a_2]$ and $[a_3, a_4]$, and apply optimal 1:2 method twice i.e., $\theta_{1:2}([a_1, a_2])$ and $\theta_{1:2}([a_3, a_4])$. We can show (proof in Appendix A.3) that the former alternative is preferable and introduces less variance, i.e.,

$$\text{Var}[\theta_{2:4}(\mathbf{a})] \leq \text{Var}[\theta_{1:2}([a_1, a_2])] + \text{Var}[\theta_{1:2}([a_3, a_4])] \tag{19}$$

### 4.4  APPROXIMATELY OPTIMAL 2:4 METHOD

As shown in Table 3, in terms of time complexity, the optimal 2:4 method might not be feasible. Using insights gained from the optimal solution, we now present a simple near-optimal 2:4 method called approx-MVUE. The idea is simple. We first remove from the block one element $a_i$, where $i$ is chosen with probability

$$p_i = \frac{a_i}{a_1 + a_2 + a_3 + a_4}.$$

In order to select a second element, we repeat the same procedure for the three remaining elements with probability

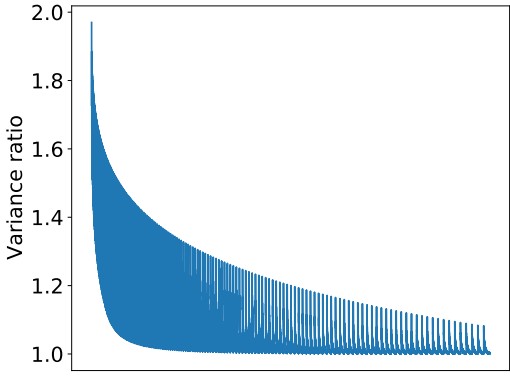

Figure 2: Ratio between the variance (Equation (12)) of the approx-MVUE 2:4 and MVUE 2:4 when scanning (step size 0.005 all possible values of a block $[a_1, a_2, a_3, 1]$, where $0 \leq a_1, a_2, a_3 \leq 1$. Notice that ratio is bounded below 2. The maximum is achieved near the left edge, when $a_4 \gg \max(a_1, a_2, a_3)$.

$$p_j = \frac{a_j}{a_1 + a_2 + a_3 + a_4 - a_i}.$$

Thus, each element is chosen with probability:

$$p_i = \frac{|a_i|}{\sum_j |a_j|} + \sum_{k \neq i} \frac{|a_k|}{\sum_j |a_j|} \frac{|a_i|}{\sum_{j \neq k} |a_j|} \tag{20}$$

The effect of this approximated method on the variance of the estimator is presented in Figure 2 by the ratio of the variance between the two methods:

$$\frac{\text{Var}(\theta_{2:4}^{\text{approx}})}{\text{Var}(\theta_{2:4}^{\text{opt}})},$$

where both variances are calculated analytically using Equation (12). Without loss of generality for a block $[a_1, a_2, a_3, a_4]$ where $a_1 \leq a_2 \leq a_3 \leq a_4$, we set $a_4 = 1$ and scan with small steps all combinations of $a_1, a_2, a_3$. The scan suggests the variance ratio is bounded below two[1], and therefore the approximate method is a 2-approximation of the old method. As can be seen in Table 3 the approximated method reduces the complexity time of MVUE 2:4 by $\sim 70\%$, in our non-optimized implementation. Moreover, in Appendix B we give additional details on this experiment and compare the number of operations required for finding MVUE mask with the computation gain achieved. Based on that we derive a simple rule to decide for each layer when neural gradient pruning is efficient.

## 5 EXPERIMENTS

In this section, we demonstrate the effectiveness of our proposed method over several vision and language models. First we show the effect of the proposed method for the fine-grained N:M structured sparsity on the neural gradients. Then we combine this method with the fine-grained N:M transposable-weights method (Hubara et al., 2021), allowing the acceleration with N:M structured sparsity in all training GEMM. Moreover, we show the combination of N:M structured sparsity in all training GEMM with 8-bit quantization achieving non or small accuracy degradation. Experimental details appear in Appendix A.4.

Table 3: Overhead of different algorithms for finding the required masks: ratio of their running time over regular training (ResNet50). Notice the overhead reduction in the Approx-MVUE 2:4 in comparison to MVUE 2:4. All experiments were run in FP32 without sparse-tensor cores and in a non-optimized implementation.

| Method | Overhead (%) |
|---|---|
| MVUE 1:2 | 1 % |
| MVUE 2:4 | 95 % |
| Approx-MVUE 2:4 | 3 % |

Notice that, while the Nvidia A100 tensor core supports sparse GEMM operation with 2:4 structured pruning, their software only supports it for inference (weights pruning) and not for training. Since there is currently no support for our method in any AI accelerator, we cannot show an actual training time reduction. Therefore, in Appendix B, we attempt to estimate when neural gradient pruning is efficient by comparing the number of operations required for finding MVUE mask with the computation gain achieved. We note, that this is the common practice in the neural network compression literature, where the algorithms often appear before the hardware that can support them. For example, though we can find FP8 training publications since 2019 (Sun et al., 2019), only recently did Nvidia announce their first GPU that supports the FP8 format (H100).

**N:M structured sparsity on the neural gradients** In Table 4 we show the results of applying the suggested N:M structured sparsity for various models and datasets. The 1:2 results refer to the MVUE method (Section 4.1) while the 2:4 results refer to the approximate-MVUE method (Section 4.4). Notice the comparison with the traditional greedy method of keeping the largest elements in each block. While the greedy method has a very significant degradation, the proposed method achieved small or no degradation with the proposed methods.

**Accelerating all training phases** In Table 5 we showed the results of the combination between the proposed N:M MVUE for the neural gradients and the N:M transposable weights presented in Hubara et al. (2021). The combination between both methods allows to be able to accelerate with N:M structured sparsity all training GEMM operations with minimal or no accuracy degradation. Moreover, in Table 6 we show the combination of N:M sparsity in all training GEMM with 8-bit quantization. For the quantization we used INT8 (Choi et al., 2018a) in the weights and activations and FP8 (Chmiel et al., 2021b) in the neural gradients.

---

[1]The largest values are near the left edge of the scan, which represents the limit where $a_4 \gg \max(a_1, a_2, a_3)$. Near this edge, we additionally checked with very small (logarithmically spaced) step sizes that the variance ratio is bounded below two.

Table 4: Effect of applying the proposed MVUE 1:2 and approx-MVUE 2:4 on the neural gradients for different models and datasets. Notice that in most cases MVUE 1:2 achieved full precision accuracy and when it did not, the approx-MVUE 2:4 method closed the gap. 'Greedy' refers to the traditional method of keeping the $N$ largest elements in each block (minimum MSE) which suffers from a significant degradation.

| Model | Dataset | FP32 | MVUE 1:2 | Approx-2:4 | Greedy |
|---|---|---|---|---|---|
| ResNet18 | ImageNet | 70.6 % | 70.58 % | 70.6 % | 48.2 % |
| ResNet50 | ImageNet | 77.2 % | 76.4 % | 77.12 % | 59.3 % |
| ResNext50 | ImageNet | 77.61 % | 76.05 % | 77.55 % | 60.7 % |
| DenseNet-121 | ImageNet | 74.4 % | 74.1 % | 74.4 % | 70.3 % |
| ViT-B16 | Cifar10 | 98.8 % | 98.4 % | 98.7 % | 96.7 % |
| Bert finetune | Squad | 79.38 (EM) 87.03 (F1) | 78.55 86.41 | 79.15 86.82 | 66.2 70.2 |
| Bert pretrain | Wiki | 0.72 (MLM) | 0.718 | 0.72 | 0.68 |
| Transformer | WMT En-De | 27.5 (BLUE) | 27.32 | 27.44 | 25.55 |

Table 5: Effect of applying N:M structured sparsity in all training phases. We combine the suggested MVUE 1:2 and approx-MVUE 2:4 for the neural gradients in the update phase with the transposable weights of Hubara et al. (2021) in the forward and backward phases.

| Model | Update ($G$) | Forward ($W$) | Backward ($W^T$) | Accuracy |
|---|---|---|---|---|
| ResNet18 | FP32 | FP32 | FP32 | 70.6 % |
| | FP32 | 2:4 | 2:4 | 70.5 % |
| | MVUE 1:2 | 2:4 | 2:4 | 70.4 % |
| | Approx-2:4 | 2:4 | 2:4 | 70.6 % |
| ResNet50 | FP32 | FP32 | FP32 | 77.2 % |
| | FP32 | 2:4 | 2:4 | 77.1 % |
| | MVUE 1:2 | 2:4 | 2:4 | 75.6 % |
| | Approx-2:4 | 2:4 | 2:4 | 77.1 % |
| ResNext50 | FP32 | FP32 | FP32 | 77.61 % |
| | FP32 | 2:4 | 2:4 | 77.4 % |
| | MVUE 1:2 | 2:4 | 2:4 | 75.88 % |
| | Approx-2:4 | 2:4 | 2:4 | 77.37 % |
| Transformer | FP32 | FP32 | FP32 | 27.5 (BLUE) |
| | FP32 | 2:4 | 2:4 | 27.5 |
| | MVUE 1:2 | 2:4 | 2:4 | 27.19 |
| | Approx-2:4 | 2:4 | 2:4 | 27.35 |

Table 6: Effect of the combination of N:M structured sparsity in all training phases with 8 bit quantization on ResNet18/50 in ImageNet datasets. For the quantization we used INT8 (Choi et al., 2018a) in the weights and activations and FP8 (Chmiel et al., 2021b) in the neural gradients.

| Model | Update ($G$) | Forward ($W$) | Backward ($W^T$) | Accuracy |
|---|---|---|---|---|
| ResNet18 | FP32 | FP32 | FP32 | 70.6 % |
| | Approx-2:4 + 8-bit | 2:4 + 8-bit | 2:4 + 8-bit | 70.3 % |
| ResNet50 | FP32 | FP32 | FP32 | 77.2 % |
| | Approx-2:4 + 8-bit | 2:4 + 8-bit | 2:4 + 8-bit | 76.48 % |

## 6 DISCUSSION

**Conclusions** In this work, we studied the effect of N:M structured sparsity on the neural gradients to accelerate the update phase. Based on a previous work (Chmiel et al., 2021b), which showed the importance of unbiasedness of the neural gradients in quantization, we suggest an unbiased minimum variance method for pruning the neural gradient using 1:2 and 2:4 structured sparsity. Since the optimal 2:4 method may not be feasible in term of complexity, we suggest an approximate method

which increases the variance only by a factor of 2 (making it a 2-approximation). We showed that our methods achieved small or no degradation while the traditional greedy method completely failed. Moreover, we combine our method with a previous method for transposable weights (Hubara et al., 2021). This enables a potential acceleration by x2 of all GEMMs in the training process using only N:M fine grained sparsity. In the following paragraphs we will discuss additional aspects of N:M structured sparsity acceleration including the benefits of pruning both matrices involved in a single matrix multiplication and the potential improvement in the inference phase (Equation (1)).

**Should we prune both matrices involved in the matrix multiplication?**   So far, fine-grained sparsity papers focused on pruning only one matrix in each phase (Nvidia, 2020; Zhou et al., 2021; Hubara et al., 2021), achieving up to x2 acceleration in the corresponding phase. Since pruning the weights is the most common approach, an interesting question is what would happen if we prune two matrices in one GEMM, such as both the weight and activation matrices in the forward phase? Can this accelerate the computation? Next, we explain why pruning both matrices cannot further accelerate computation (i.e., by x4) in modern accelerators, but that it reduces the required bandwidth and thus simplify the hardware design.
We start by analyzing what is the expected acceleration when both matrices (that are involved in the matrix multiplication) follow N:M fine-grained sparsity. Assuming we have two N:M fine-grained blocks $b_W$ and $b_H$ with masks $M_{b_W}$ and $M_{b_H}$, the number of Multiply and Accumulate operations (MACs) required for multiplying and accumulating the blocks may vary from zero to N. For example, for 2:4 fine-grained sparsity, there are $\binom{4}{2} = 6$ possible mask configurations. Thus, the expected number of MACs in a block, assuming uniform distribution on the non-zeros in the blocks, would be:

$$\mathbb{E}[\#\text{MACs}(b_H, b_W)] = \mathbb{E}\left[\mathbb{E}[\#\text{MACs}(b_H, b_W)|M_{b_H}]\right] = \frac{1}{6} \cdot 0 + \frac{1}{6} \cdot 2 + \frac{4}{6} \cdot 1 = 1 \qquad (21)$$

Thus on average, for each block we get $N/2$ MACs. While some architectures (such as CPU) can avoid all unnecessary multiplications, architectures with a systolic array at the core of their matrix multiplication engine, as modern hardware accelerators, must always assume the worst case. Therefore, for these types of architectures, we cannot achieve an additional compute reduction by pruning both matrices involved in the matrix multiplication. Yet, the bandwidth reduction for both matrices is the same. This property helps support sparse and dense matrix multiplication without creating dedicated hardware which has twice the bandwidth to one of the matrices involve in the GEMM. It is specifically important when targeting higher sparsity. For instance, if only the activations obey 1:4 fine-grained structure then the weights bandwidth is x4 higher than the activations bandwidth as for every single block we bring one activations and four weights to the engine. In summary, pruning both matrices cannot accelerate modern accelerator computation, but reduces the required bandwidth, and thus simplifies the hardware design. Next, we explain how to use such a scheme for inference acceleration by pruning both weights and activations.

**Inference acceleration with activation pruning**   Can we accelerate the network using N:M sparsity on additional tensors? So far, fine-grained sparsity was applied in the forward pass (Equation (1)) only to the weight matrix. Next, we first show the effect of pruning the activations and then we combine both weights and activations pruning to improve the inference phase.
In Appendix A.5 we demonstrate that one can also accelerate inference by applying N:M sparsity on the activations. Specifically, in Appendix Table 7 we experimented with greedy N:M fine-grained sparse activations on ResNet18 and ResNet50 over ImageNet dataset, wherein for each block of size M we keep the M-N larger elements. Note that in CNNs the activations memory footprint is much larger than the weights footprint (especially for the first set of layers), so in term of memory reduction activations pruning is more effective than weights pruning. Throughout our experiments, we did not change the training regime and pruned the activations from scratch. As can be seen, applying only fine-grained sparse activations results in notable accuracy degradation. However, a simple fix is to apply ReLU before the fine-grained sparse activations; this results in on-par accuracy for both ResNet18 and ResNet50. In Appendix Table 8 we experimented with fine-grained N:M structured sparsity both weights and activations. To compete with the latest inference acceleration results based on quantization-aware techniques, we further quantize the weights and activation to 4-bit and show better results in terms of bit-operations (BOPS) (Wang et al., 2020) to accuracy than 2-bits inference methods. Notably, While using 2-bit for both weights and activations has the potential of x4 acceleration, in modern hardware it would probably be only up to x2 as is simplifies the design and reduces the die area, see Nvi where 16bit GEMM has 800 TFLOPS and 8bit GEMM has 1600 TFLOPS). Thus we argue that our 4-bit with sparse weights and activations has a similar potential.

## ACKNOWLEDGEMENT

The research of DS was Funded by the European Union (ERC, A-B-C-Deep, 101039436). Views and opinions expressed are however those of the author only and do not necessarily reflect those of the European Union or the European Research Council Executive Agency (ERCEA). Neither the European Union nor the granting authority can be held responsible for them. DS also acknowledges the support of Schmidt Career Advancement Chair in AI.

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
