# OpenReview forum: "Minimum Variance Unbiased N:M Sparsity for the Neural Gradients"
_ICLR.cc/2023/Conference — ICLR 2023 notable top 25%_

### Official Review · Reviewer_ccBg · 2022-10-24

**Confidence:** 3
**Correctness:** 3
**Technical Novelty And Significance:** 3
**Empirical Novelty And Significance:** 3
**Recommendation:** 8

**Clarity, Quality, Novelty And Reproducibility:**

The paper is well written and the results are presented in a clear way.
Proposed pruning scheme seems straightforward to implement, but having access to a reference implementation may still be valuable.

**Strength And Weaknesses:**

The paper is well written and proposes very attractive improvements to training speed without affecting the accuracy.
Given a relatively general title, I'd be curious to see some results on using the approximate method in setups with different sparsity setups, e.g, 4:8, and also higher sparsity, e.g., 1:4.

**Summary Of The Paper:**

The paper aims to accelerate training by pruning the calculated gradients. More specifically, the authors propose 1:2 and 2:4 minimum-variance unbiased estimators and show their effectiveness on a number of tasks, where accuracy is not severely affected by the proposed pruning scheme.

**Summary Of The Review:**

A well written paper with interesting and relevant results for practical applications of DL.

---

> ### Author Response · Authors · 2022-11-10
> **Response to reviewer ccBg**
>
> $\textbf{Q1}$:  "I'd be curious to see some results on using the approximate method in setups with different sparsity setups, e.g, 4:8, and also higher sparsity, e.g., 1:4."
>
> $\textbf{A1}$: The paper focused on 2:4 sparsity as it is currently supported is hardware. Yet, following reviewer's request, we added in the appendix B.4 additional results for 4:8 and 1:4. For 4:8 we get 70.6 \% and 77.15\% in ResNet18 and ResNet50 respectively, while for 1:4 we get 69.93\% and 75.8\% in ResNet18 and ResNet50  respectively.

---

### Official Review · Reviewer_B9mF · 2022-10-25

**Confidence:** 4
**Correctness:** 4
**Technical Novelty And Significance:** 3
**Empirical Novelty And Significance:** Not applicable
**Recommendation:** 8

**Clarity, Quality, Novelty And Reproducibility:**

Besides the above suggestion, the paper is generally easy to follow.

The results demonstrate the proposed 2x pruning procedure doesn't hurt performance significantly.

The proposed pruning operator is, to my knowledge, novel and seems simple to reproduce from the provided equations.


**Strength And Weaknesses:**

# Strengths

The paper provides a novel method for pruning the back-propagation gradients.

The proposed method does not hurt network performance, whereas naive greedy baselines hurt performance significantly.

The paper is well-written and reasonably easy to follow.

The paper provides a significant discussion of the implications the paper's results have on hardware acceleration and design

# Weaknesses
The paper doesn't make a strong case for why the purning operator needs to be unbiased and only points vaguely to Chmiel et al.'s finding that "for the neural gradients... it is critical to use unbiased quantization (i.e., Bias[θ(a)] = 0)."

For completely understandable reasons (no hardware support for pruning during training), the paper is unable to demonstrate any actual training time reductions. Hopefully this changes as new hardware is developed.

# Suggestion
The introduction of the MSE of the pruning operation describes $a$ as scalar, rather than a vector. The resulting discussion is then confusing as there's no stochasticity in $a$ and one is left asking why not just let $\theta(a)=a$. I suggest introducing $a$ as a vector from the start, as this will make the resulting discussion easier to follow -- there are many choices for the pruning operator which would produce different MSEs.


**Summary Of The Paper:**

This paper develops a new scheme for pruning the gradients passed through a network during back-propagation. In particular, the proposed method designs a stochastic pruning operator which is unbiased and provides the minimum variance output. Both 1:2 (2 inputs, 1 non-zero output) and 2:4 (4 inputs, 2 non-zero outputs) variants of this operator were developed. In testing, networks pruned with the proposed operator demonstrated little-to-no degradation in performance across a range of network architectures and classification problems.


**Summary Of The Review:**

The paper provides a simple gradient pruning procedure that could accelerate future hardware systems. While the form of the pruning operator could use further motivation, I'm overall in favor of this paper's publication.

---

> ### Author Response · Authors · 2022-11-10
> **Response to reviewer B9mF**
>
> $\textbf{Q1}$: "The paper doesn't make a strong case for why the purning operator needs to be unbiased and only points vaguely to Chmiel et al.'s finding that "for the neural gradients... it is critical to use unbiased quantization (i.e., Bias[$\theta$(a)] = 0)."
>
> $\textbf{A1}$: Excellent question! As mentioned in section 3.2 in Chmiel et al [A], one of the textbooks assumptions (e.g. Bottou et al [B]) required for SGD convergence is that the expectation of the weight mini-batch gradient is sufficiently close to the weight full batch gradient. Chmiel et al [A] showed that unbiasedness in the neural gradient leads to an unbiased estimator of the weight mini-batch gradient, which enables proper convergence of SGD. We added this explanation in section 3 of the revised paper.
>
> $\textbf{Q2}$: "I suggest introducing $a$ as a vector from the start, as this will make the resulting discussion easier to follow -- there are many choices for the pruning operator which would produce different MSEs."
>
> $\textbf{A2}$: Great idea. We changed it in the revised version in section 3.
>
> ==============
>
> [A] Brian Chmiel, Ron Banner, Elad Hoffer, Hilla Ben Yaacov, and Daniel Soudry. Logarithmic unbiased
> quantization: Simple 4-bit training in deep learning. ArXiv, abs/2112.10769, 2021a.
>
> [B] ] Léon Bottou, Frank E Curtis, and Jorge Nocedal. Optimization methods for large-scale machine
> learning. Siam Review, 60(2):223–311, 2018.

---

### Official Review · Reviewer_e1dv · 2022-10-28

**Confidence:** 4
**Correctness:** 4
**Technical Novelty And Significance:** 3
**Empirical Novelty And Significance:** 2
**Recommendation:** 8

**Clarity, Quality, Novelty And Reproducibility:**

The paper is well written with clear analyses and provides code for the proposed method.

**Strength And Weaknesses:**

Strengths
------------

- The theoretical analysis is well written and intuitive
- Accelerating GEMMs is a very important topic of research as it affects all modern neural network architectures
- Thorough experimental evaluations show that the method indeed does not hurt performance and can be used to train a variety of models

Weaknesses
--------------

- The main weakness of the paper is the lack of real world performance improvements due to lack of custom kernels or support from currently available accelerators. The speedup that is mentioned is a best case scenario and makes the method seem more useful than it is probably.
- Another weakness is the lack of comparisons with MSE based sparsity for the gradients. Even though previous work has shown that unbiased estimators are more important for the gradients than the weights, it would strengthen the paper significantly if instead of the greedy estimator the paper also compared with MSE optimal biased estimators.

Typos
-------

- eq. 8 typo it should be $sign(a_2) \left(\|a_1\| + \|a_2\|\right)$
- Fig. 2 typo $a_4 >> \max(a_1, a_2, a_3)$

**Summary Of The Paper:**

The paper presents a method for a stochastic unbiased masking of the gradients such that half of them are 0 and they can be used to accelerate the matrix multiplications in an accelerator such as a GPU. In particular, the method induces a form of sparsity called N:M sparsity where N out of M consecutive (in memory) elements are 0. The authors propose to use a masking such that the gradients are still unbiased but also such that they have the minimum variance. Thorough experiments show that using their approximate 2:4 sparsity algorithm as well as the exact 1:2 algorithm, allows training a variety of neural network architectures on both images and text with minimal loss in final performance, if any. Moreover, the authors show that their algorithm can be combined with similar algorithms for the forward and backward pass as well as quantization without significant drop in performance.

**Summary Of The Review:**

The method is very intuitive and straightforward. It may be slightly incremental work, however I believe it is a clear contribution.

---

> ### Author Response · Authors · 2022-11-10
> **Response to reviewer e1dv**
>
> $\textbf{Q1}$: "The main weakness of the paper is the lack of real world performance improvements due to lack of custom kernels or support from currently available accelerators. The speedup that is mentioned is a best case scenario and makes the method seem more useful than it is probably."
>
> $\textbf{A1}$: Indeed, there is no hardware support for the proposed method, and so we are not able to show real world performance. However we analyze this method theoretically in Appendix B. As mentioned in section 5, this is a common practice in the neural networks acceleration literature,  where algorithms often appear before the hardware that can support them (since without a working algorithm, there is no reason to invest in the hardware to support a new functionality).
>
> $\textbf{Q2}$: "lack of comparisons with MSE based sparsity for the gradients. Even though previous work has shown that unbiased estimators are more important for the gradients than the weights, it would strengthen the paper significantly if instead of the greedy estimator, the paper also compared with MSE optimal biased estimators."
>
> $\textbf{A2}$: The 'greedy' method *$\textbf{is}$* the MSE optimal biased estimator: the most common pruning method is zeroing the smallest elements (in term of magnitude) for each block, which minimize the MSE under predefined sparsity level. We clarified it in the updated version in page 4: "Notice, the proposed MVUE method has the best accuracy, although it does not minimize the MSE, as done by the `greedy' method.", and also in the caption of table 2.
>
>
> $\textbf{Q3}$: Typos
>
> $\textbf{A3}$: We fix the typos in the updated version

---

### Author Response · Authors · 2022-11-10
**General comment**

We thank all the reviewers for their detailed feedback, and for expressing positive opinions: Reviewer e1dv ---$\textit{"I believe it is a clear contribution.."}$; Reviewer B9mF --- $\textit{"The paper is well-written and reasonably easy to follow.."}$; and Reviewer ccBg --- $\textit{"A well written paper with interesting and relevant results for practical applications of DL."}$.

We uploaded a new revision of the paper and supplementary material to address all remarks. All the changes are marked in red. For additional details, see the answer for each reviewer concerns. Please let us know if there are any additional comments.

---

### Decision · Program_Chairs · 2023-01-20

**Decision:**

Accept: notable-top-25%

**Justification For Why Not Higher Score:**

The paper is unable to demonstrate any actual training time reductions.

**Justification For Why Not Lower Score:**

The proposed method is novel and is supported by extensive experiments, and is of importance for the ICLR community.

**Metareview: Summary, Strengths And Weaknesses:**

The paper investigates application of N:M sparsity for the neural gradients (i.e. loss gradients with respect to the intermediate neural layer outputs), by using a masking mechanism such that the gradients are still unbiased and have minimum variance. Experimental results confirm that using the proposed approximate 2:4 and the exact 1:2 algorithms, allows training a variety of neural network architectures on both images and text with minimal loss in final performance. The proposed idea is novel and the experimental results are convincing. The main downside is that the paper is unable to demonstrate any actual training time reductions, as existing hardwares do not support N:M sparsity for gradients. Still, the proposed method is valuable and as such hardwares are expected to be developed in near future. Hence, I recommend acceptance.

**Note From Pc:**

if the above contains the word "oral" or "spotlight" please see: "oral" presentation means -> notable-top-5% and "spotlight" means -> notable-top-25%. As stated in our emails, we are disassociating presentation type from AC recommendations